# Trajectory Tracking Control for Robotic Manipulator Based on Soft Actor–Critic and Generative Adversarial Imitation Learning

**DOI:** 10.3390/biomimetics9120779

**Published:** 2024-12-21

**Authors:** Jintao Hu, Fujie Wang, Xing Li, Yi Qin, Fang Guo, Ming Jiang

**Affiliations:** School of Computer Science and Technology, Dongguan University of Technology, Dongguan 523808, China; 221115202@dgut.edu.cn (J.H.); lixing8245@163.com (X.L.); qinyidee@163.com (Y.Q.); 2019091@dgut.edu.cn (F.G.); jiangm@dgut.edu.cn (M.J.)

**Keywords:** generative adversarial imitation learning, end effector, robotics tracking control, long short-term memory, deep reinforcement learning

## Abstract

In this paper, a deep reinforcement learning (DRL) approach based on generative adversarial imitation learning (GAIL) and long short-term memory (LSTM) is proposed to resolve tracking control problems for robotic manipulators with saturation constraints and random disturbances, without learning the dynamic and kinematic model of the manipulator. Specifically, it limits the torque and joint angle to a certain range. Firstly, in order to cope with the instability problem during training and obtain a stability policy, soft actor–critic (SAC) and LSTM are combined. The changing trends of joint position over time are more comprehensively captured and understood by employing an LSTM architecture designed for robotic manipulator systems, thereby reducing instability during the training of robotic manipulators for tracking control tasks. Secondly, the obtained policy by SAC-LSTM is used as expert data for GAIL to learn a better control policy. This SAC-LSTM-GAIL (SL-GAIL) algorithm does not need to spend time exploring unknown environments and directly learns the control strategy from stable expert data. Finally, it is demonstrated by the simulation results that the end effector of the robot tracking task is effectively accomplished by the proposed SL-GAIL algorithm, and more superior stability is exhibited in a test environment with interference compared with other algorithms.

## 1. Introduction

With artificial intelligence technology advancing and the concept of Industry 4.0 emerging, the development of robot manipulators has gained significant attention due to their numerous applications in various sectors [1]. The operational scenarios are no longer fixed, predefined, or well known [2]. Regarding trajectory tracking control for a end effector of a manipulator, it is essential for each position to track a desired trajectory closely [3]. Various methods are proposed to achieve satisfactory tracking control effects, including PID control, sliding-mode control, adaptive tracking control, and model predictive control [4,5,6,7,8]. However, many of the mentioned methods necessitate a precise model of the robot, and complex controllers often need more control parameters. Fortunately, neural networks demonstrate exceptional approximation capabilities for uncertain mathematical models and are among the effective methods for addressing nonlinear system control problems.

In [9], an adaptive asymptotic prescribed performance approach using a radial basis function neural network (RBFNN) is proposed for a hydraulic manipulator. An end-to-end (E2E) deep learning method is proposed for robot classification and real-time motion control in [10]. Through forward synchronous learning and adjustable Rectified Linear Unit (ReLU), the stability and robustness of the deep learning algorithm are ensured. However, neural network training for tracking control tasks in robotic manipulators is susceptible to local optimal problems. It is worth mentioning that the appearance of DRL expands the options for designing control algorithms [11]. The DRL collects experience data through interactive learning of the agent with the environment, utilizing these data to optimize the loss function and train the model. Consequently, local optimal solution problems can be flexibly handled.

In [12], the fuzzy Q-learning network, a typical DRL scheme, is used to resolve the problem of trajectory tracking of an Uncertain Quadrotor System (UQS). The state–action–reward–state–action (SARSA) algorithm is used to realize the positioning of random target and fixed target points of the 3-DoF manipulator end effector in [13]. While Q-learning and SARSA are effective for addressing typical reinforcement learning (RL) tasks, they are not suitable for continuous spaces. It should be noted that trajectory tracking control problems typically involve continuous state spaces. Thus, in order to complete the tracking control task of pneumatic musculoskeletal robots, a model-based RL (MBRL) method is proposed in [14]. In [15], a model-based offline RL method is introduced to enhance the control performance, which is combined with a torque controller to implement tracking control for a manipulator. However, learning an effective control policy is more challenging with model-based RL (MBRL) compared with model-free RL (MFRL). The MFRL algorithms are not dependent on specific environment models, which enables them to perform control tasks effectively in dynamic and uncertain environments. In [16], a distributed Proximal Policy Optimization (PPO) algorithm [17] based on the LSTM network is proposed to train robotic arms and mobile robots to track a given trajectory. In order to control a 2-DoF manipulator with an unknown deadzone to track a desired trajectory, an actor–critic RL method is adopted in [18]. An end-to-end target tracking method utilizing a DRL approach is proposed to address the complexity of control in free-floating space manipulators (FFSMs) in [19]. The challenge of locating and tracking the eddy center in an uncharted environment using an Underactuated Autonomous Surface and Underwater Vehicle (UASUV) is explored in [20], in which the SAC algorithm and LSTM are combined. While MFRL is extensively studied in tracking control tasks, most existing algorithms necessitate numerous training epochs to derive reasonable control policies.

Fortunately, the proposal of GAIL solves the problem of time-consuming training in many tasks [21]. GAIL learns the optimal policy directly from expert data by combining Inverse RL (IRL) [22] with Generative Adversarial Networks (GANs) [23]. In [24], a GAIL-based DRL method is introduced to train an Autonomous Underwater Vehicle (AUV) to emulate expert paths. Similarly, in [25], a GAIL-driven navigation system for Unmanned Surface Vehicles (USVs) is proposed, which learns a policy to mimic expert trajectories. The above-mentioned references [24,25] use the on-policy algorithms PPO and Trust Region policy optimization (TRPO) [26] as generators, respectively [27]. Although the on-policy algorithm is stable, it is challenging to learn a superior control policy due to the expense of sample efficiency GAIL policy training. When the robot environment model is unknown, the exploration ability of the algorithm is crucial. Meanwhile, off-policy methods exhibit superior sample efficiency, but their training process is unstable [28]. Fortunately, by utilizing the LSTM network, the evolving patterns of joint positions over time are competently captured and comprehended, thereby mitigating instability during the training process of robotic manipulators for tracking control tasks. Therefore, the main focus of this paper is to embed LSTM into the SAC algorithm and utilize the resulting SAC-LSTM algorithm as a generator for GAIL training. This approach aims to address the trajectory tracking control problem for the end effector of manipulator systems under input saturation and random interference. The principal contributions are summarized as follows:

(1) Inspired by [20], the LSTM is introduced to enhance the stability of the end effector of the manipulator when tracking the target trajectory. During trajectory tracking control, the sequential states of the manipulator are inputted into the LSTM layer to produce hidden states. These hidden states are processed to generate Gaussian action parameters. This method effectively captures sequential dependencies, thereby improving the robustness and adaptability of trajectory tracking control.

(2) In the above-mentioned references [24,25], the on-policy algorithm is selected as the generator of GAIL. In this paper, the off-policy algorithm SAC combining LSTM is chosen as a generator, and the policies trained by SAC-LSTM are selected as the expert data. Through GAIL training, the agent learns control strategies directly from expert data and is able to quickly learn strategies that are superior to expert data.

(3) A SL-GAIL algorithm is proposed by combining the SAC-LSTM and GAIL method. This algorithm is used to train the robot to track the target trajectory. This approach reduces unnecessary exploration, accelerates the acquisition of superior control strategies, and helps improve the efficiency and robustness of trajectory tracking for the end effector of the robotic manipulator.

The controlled objects and control methods used in the above literature are shown in Table 1.

## 2. System Description

### 2.1. Dynamics Model

The dynamical equations of the system can be obtained by deriving the equations from the Lagrangian functions. The dynamic equation for a *n*-DOF manipulator is described as follows [29]:(1)$$\begin{array}{c} \left.\begin{array}{c} M\left(\theta \right)\ddot{\theta }+C\left(\theta ,\dot{\theta }\right)\dot{\theta }+G\left(\theta \right)=\tau +\tau_{d} \end{array}\right. \end{array}$$

Among them, θ, $\dot{\theta }$, and $\ddot{\theta }$∈$R^{n}$ represent the joint position, joint velocity, and joint acceleration, respectively. M(θ)∈$R^{n\times n}$ expresses the inertia matrix, $C(\theta ,\dot{\theta })$∈$R^{n\times n}$ represents the centripetal and Coriolis torque, G(θ)∈$R^{n}$ is the gravitational force, τ∈$R^{n}$ is the to the joint torques, and $\tau_{d}\in R^{n}$ is a random interference. In this paper, the Phantom Omni robot is studied. The robot is shown in Figure 1a. The schematic diagram of the Phantom Omni is shown in Figure 1b, in which the reference frames used in dynamics are outlined [30]. The robot is a 3-DoF manipulator and is modeled by Equation (Equation 1), and the *M*, *C*, and *G* matrices are as follow [31]:$$\begin{array}{cc} M(\theta ) & =\left[\begin{array}{ccc} m_{11} & 0 & 0\\ 0 & m_{22} & m_{23}\\ 0 & m_{32} & m_{33} \end{array}\right],G\left(\theta \right)=\left[\begin{array}{c} 0\\ gk_{5}c_{2}+gk_{6}c_{23}\\ gk_{6}c_{23} \end{array}\right],\\ C(\theta ,\dot{\theta }) & =\left[\begin{array}{ccc} -a_{1}{\dot{\theta }}_{2} & -a_{1}{\dot{\theta }}_{1} & -a_{2}{\dot{\theta }}_{1}\\ a_{1}{\dot{\theta }}_{1} & -a_{3}{\dot{\theta }}_{3} & -a_{3}\left({\dot{\theta }}_{2}+{\dot{\theta }}_{3}\right)\\ a_{2}{\dot{\theta }}_{1} & a_{3}{\dot{\theta }}_{2} & 0 \end{array}\right] \end{array}$$
where $m_{11}=k_{1}+k_{2}c_{2}^{2}+k_{3}c_{23}^{2}+2k_{4}c_{2}c_{23},m_{22}=k_{2}+k_{3}+2k_{4}c_{3},m_{23}=k_{3}+k_{4}c_{3},$
$m_{32}=m_{23},m_{33}=k_{3},a_{1}=k_{2}c_{2}s_{2}+k_{3}c_{23}s_{23}+k_{4}c_{2\times 23},a_{2}=k_{3}c_{23}s_{23}+k_{4}c_{2}s_{23},$
$a_{3}=k_{4}s_{3},s_{i}=sin\left(\theta_{i}\right),s_{23}=\sin\left(\theta_{2}+\theta_{3}\right),c_{i}=\cos\left(\theta_{i}\right),c_{23}=\cos\left(\theta_{2}+\theta_{3}\right),c_{2\times 23}=\cos\left(2\theta_{2}+\theta_{3}\right)$.

### 2.2. Kinematics Model

The real-time joint angle and joint velocity of the manipulator are obtained by solving Equation (Equation 1). Thus, the relationships between the end effector position vector and joint space vector are expressed as [32]
(2)$$\begin{array}{cc} x & =f\left(\theta \right) \end{array}$$
(3)$$\dot{x}=J\left(\theta \right)\dot{\theta }$$
where $f\left(\cdot \right):R^{n}\to R^{n}$ is the mapping between the joint space and task space. $J\left(\theta \right)\in R^{n\times n}$ is the Jacobian matrix. According to forward kinematics, the Cartesian coordinates of the end effector are expressed as follows [32]:(4)$$x=\left[\begin{array}{c} x\\ y\\ z \end{array}\right]=\left[\begin{array}{c} \left(l_{2}\cos\left(\theta_{2}\right)+l_{3}\cos\left(\theta_{2}+\theta_{3}\right)\right)\cos\left(\theta_{1}\right)\\ \left(l_{2}\cos\left(\theta_{2}\right)+l_{3}\cos\left(\theta_{2}+\theta_{3}\right)\right)\sin\left(\theta_{1}\right)\\ l_{2}\sin\left(\theta_{2}\right)+l_{3}\sin\left(\theta_{2}+\theta_{3}\right) \end{array}\right]$$
the Jacobian matrix, *J*, is expressed [33]
(5)$$J=\left[\begin{array}{ccc} -rs_{1} & -zc_{1} & -l_{3}c_{1}s_{23}\\ rc_{1} & -zs_{1} & -l_{3}s_{1}s_{23}\\ 0 & r & l_{3}c_{23} \end{array}\right]$$
where $r=\sqrt{x^{2}+y^{2}}$. According to Equations (Equation 3) and (Equation 5), the $\dot{x}$ is as follows:(6)$$\dot{x}=\left[\begin{array}{c} \dot{x}\\ \dot{y}\\ \dot{z} \end{array}\right]=\left[\begin{array}{ccc} -rs_{1} & -zc_{1} & -l_{3}c_{1}s_{23}\\ rc_{1} & -zs_{1} & -l_{3}s_{1}s_{23}\\ 0 & r & l_{3}c_{23} \end{array}\right]\left[\begin{array}{c} \dot{\theta_{1}}\\ \dot{\theta_{2}}\\ \dot{\theta_{3}} \end{array}\right]$$

### 2.3. Control Objective

The kinematics of a robotic manipulator is generally nonlinear, and the parameters of the manipulator system are not completely known, which results in modeling uncertainties. In addition, this paper addresses the task of controlling the joint angles by applying torques to the robot arm’s joints, subsequently enabling the end effector to track the time-varying curve in the task space, which adds to the complexity and difficulty of the control process. In this paper, the control goal is designing a RL trajectory tracking controller that aims to solve the time-varying curve tracking control problem in the task space of a manipulator with saturation constraints on joint angle and torque inputs.

## 3. Preliminaries

### 3.1. Reinforcement Learning

An RL problem is typically represented as a Markov Decision Process (MDP). An MDP is defined by a tuple *D* = (*S*, *A*, *P*, *R*), where *S* denotes a set of states that the agent might encounter, *A* signifies a set of actions that the agent chooses from at any given state, *P* defines the transition dynamics, specifying the probability of moving from one state, *s*, to another state, $s^{\prime }$, given a specific action, and *R* assigns rewards for actions taken in a state and transitioning to another state.

An RL agent aims to study a policy that maps states to actions based on interactions with its environment. This learning process is illustrated in Figure 2. The agent perceives its current state, and chooses and executes an action based on its policy at each time step. Then the environment transitions to the next state according to the action, and provides a reward. The policy is updated based on the received rewards to maximize the long-term cumulative reward. The main goal is for the agent to train an optimal control policy that maximizes the cumulative discounted reward [11].

When an RL task conforms to the characteristics of an MDP, the optimization of the task can be approached by resolving the Bellman optimality equations. The optimal state value function, $V^{*}\left(s\right)$, which maximizes the expected cumulative rewards, satisfies the following Bellman optimality equation [34]:(7)$$V^{*}\left(s\right)=\sum_{a}\pi \left(s,a\right)\sum_{s^{\prime }}P_{ss^{\prime }}^{a}\left[R_{ss^{\prime }}^{a}+\gamma V^{*}\left(s^{\prime }\right)\right]$$

The Bellman optimality equation for the state-action value function, $Q^{*}\left(s,a\right)$, is given by:(8)$$Q^{*}\left(s,a\right)=\sum_{s^{\prime }}P_{ss^{\prime }}^{a}\left[R_{ss^{\prime }}^{a}+\gamma max_{a}Q^{*}\left(s^{\prime },a^{\prime }\right)\right].$$

By learning the optimal value function, an agent is able to determine the optimal policy by choosing actions that maximize this value function.

### 3.2. Soft Actor–Critic Algorithm

SAC is the most representative algorithm for actor–critic architecture and has good sampling efficiency. The goal of an ordinary RL algorithm is to maximize the expected return $\sum_{t}^{T}E_{(s_{t},a_{t})∼\pi }\left[r\left(s_{t},a_{t}\right)\right]$, while SAC is based on the maximum entropy RL theory, which introduces expectations of the policy as the overall optimization goal based on the original RL return [35]:(9)$$\pi^{*}=\underset{\pi }{\mathrm{argmax}}\sum t^{T}E_{(s_{t},a_{t})∼\pi }\left[r\left(s_{t},a_{t}\right)+\alpha H\left(\pi \left(\cdot |s_{t}\right)\right)\right]$$
where $H\left(\pi \left(\cdot |s_{t}\right)\right)=-\sum_{a_{t}}\pi \left(a_{t}|s_{t}\right)\log\pi \left(a_{t}|s_{t}\right)$ represents the policy entropy in state $s_{t}$, and α≥0 is a temperature parameter that balances the entropy term against the reward.

SAC utilizes five neural networks: two soft Q-value networks, two target Q-value networks, and a policy network. The soft Q-value network updates follow Equation (Equation 8). For a given Q-value network parameterized by $\theta_{i}$, the loss function is:(10)$$L_{Q}\left(\theta_{i}\right)=E_{(s,a,r,s^{\prime },d)∼D}\left[\frac{1}{2}{\left(Q_{\theta_{i}}\left(s,a\right)-y\left(r,s^{\prime },d\right)\right)}^{2}\right]$$
where $y(r,s^{\prime },d)$ is described as:(11)$$y\left(r,s^{\prime },d\right)=r+\gamma \left(\min_{i=1,2}Q_{\theta_{i}^{\prime }}\left(s^{\prime },{\tilde{a}}^{\prime }\right)-\alpha \log\pi_{\phi }\left({\tilde{a}}^{\prime }|s^{\prime }\right)\right)$$

SAC employs the smaller value from the two target Q-value networks to ensure stable training. The Q-value networks are updated using stochastic gradient descent:(12)$$\nabla_{\theta_{i}}L\left(\theta_{i}\right)=\nabla_{\theta_{i}}Q_{\theta_{i}}\left(s,a\right)\left[{\left(Q_{\theta_{i}}\left(s,a\right)-y\left(r,s^{\prime },d\right)\right)}^{2}\right]$$

For policy network $\pi_{\phi }\left(a|s\right)$, which can be updated directly by maximizing (minimizing negative values), the Q value and policy corresponding to the current policy network:(13)$$J_{\pi }\left(\phi \right)=E_{(s)∼D}\left[E_{a∼\pi_{\phi }}\left[\alpha \log\pi_{\phi }\left(a|s\right)-Q_{\theta_{i}}\left(s,a\right)\right]\right]$$

SAC uses the reparameterization trick to sample actions $a=f_{\phi }\left(\xi_{a};s\right)$, where $\xi_{a}\in R^{\dim(A)}$ is drawn from a fixed Gaussian distribution, and $f_{\phi }$ represents the reparameterized policy network. The policy network is then updated using stochastic gradient descent:(14)$$\begin{array}{c} \nabla_{\phi }J_{\pi }\left(\phi \right)=\nabla_{\phi }\alpha \log\pi_{\phi }\left(a|s\right)+(\nabla_{\phi }\alpha \log\pi_{\phi }\left(a|s\right)\\ -\nabla Q_{\theta_{i}}\left(s,a\right))\nabla_{\phi }f_{\phi }\left(\xi_{a};s\right) \end{array}$$

### 3.3. Long Short-Term Memory

Each cell in an LSTM network is composed of three integral components: the forget gate, the input gate, and the output gate. The forget gate controls which information from the previous cell state should be discarded. It is defined as follows: [36]:(15)$$f_{t}=\sigma \left(W_{f}\cdot \left[h_{t-1},x_{t}\right]+b_{f}\right)$$

The input gate decides which values are updated and stored in the cell state. This process is broken down into two parts: a candidate value, $\hat{C_{t}}$, is created using the tanh function; and the input gate, $\iota_{t}$, controls the extent to which this candidate value is incorporated into the cell state:(16)$$\begin{array}{c} \iota_{t}=\sigma \left(W_{i}\cdot \left[h_{t-1},x_{t}\right]+b_{i}\right)\\ {\tilde{C}}_{t}=\tanh\left(W_{C}\cdot \left[h_{t-1},x_{t}\right]+b_{C}\right) \end{array}$$
where the final component is the output gate, which determines the output of the LSTM cell. This gate combines the current cell state and the output gate activation to produce the final hidden state:(17)$$\begin{array}{c} o_{t}=\sigma \left(W_{o}\cdot \left[h_{t-1},x_{t}\right]+b_{o}\right)\\ h_{t}=o_{t}\cdot \tanh\left(C_{t}\right) \end{array}$$

The mention above σ is sigmoid function, *W* and *b* represent the weight matrix and bias term, $h_{t-1}$ is the previous hidden state, $x_{t}$ is the current input, [·] denotes the concatenation operation, $\hat{C_{t}}$ is the candidate cell state, and $C_{t}$ is the cell state. The interaction of these gates enables the LSTM unit to capture long-term dependencies, making it effective for sequential data processing.

### 3.4. Generative Adversarial Imitation Learning

GAIL enhances traditional imitation learning by incorporating principles from maximum entropy inverse reinforcement learning. It combines imitation learning with Generative Adversarial Networks (GANs) to directly learn the optimal policy that makes the distribution of the state–action pairs of the agent closely match those of expert trajectories. In GAIL, the role of the discriminator is to differentiate between the state–action pairs generated by the agent and those from the expert. Meanwhile, the generator, which represents the policy of the agent, is trained to produce state–action pairs that the discriminator classifies as expert-like. This adversarial process aims to find a saddle point (π, *D*) in the following objective function [37]:(18)$$E_{\pi }\left[\log D\left(\left(s,a\right)\right)\right]+E_{\pi_{E}}\left[\log\left(1-D\left(s,a\right)\right)\right]-\lambda H\left(\pi \right)$$
where D(s,a) represents the discriminator in GAIL. $E_{\pi }\left[\log D\left(\left(s,a\right)\right)\right]$ is the expectation over the policy of the agent, encouraging the generator to produce state–action pairs that the discriminator classifies as expert-like. $E_{\pi_{E}}\left[\log\left(1-D\left(s,a\right)\right)\right]$ is the expectation over the policy of the expert, ensuring that the discriminator correctly identifies expert state–action pairs. $H\left(\pi \right)≜E_{\pi }\left[-\log\pi \left(a|s\right)\right]$ denotes the entropy of the policy, π, which encourages exploration and prevents the policy from collapsing to deterministic behaviors. λ is the weighting parameter that determines the significance of the entropy term in the overall objective.

## 4. Control Design

### 4.1. Design of System Input/Output

In deep RL, the actor network determines an output action based on the current input state. For the trajectory tracking in the task space of the manipulator, the joint angles of the manipulator and the position of the end effector are crucial. The traditional state space includes joint angles and joint angular velocities, which can describe the internal state of the robot, but cannot directly reflect the relationship between the performance of the end effector of the robot and the target. Therefore, adding the mapped end effector position and velocity, as well as the target position and velocity, can provide more relevant information about the end effector and target trajectory, thereby helping the reinforcement learning algorithm to control more accurately. Furthermore, to guarantee that the control policy is able to obtain effective tracking control, there must be error information for trajectory tracking in the state space [38]. In this paper, the primary objective of the controller is to determine the appropriate actions according to the real-time status information of the robotic manipulator to achieve effective tracking control. The state space is described as follows:(19)$$s={\left[\begin{array}{cccccccc} \theta & \dot{\theta } & x & \dot{x} & x_{d} & \dot{x_{d}} & e & \dot{e} \end{array}\right]}^{T}$$
where *x* and $\dot{x}$ are are the end position and the velocity at the end position, respectively. $x_{d}$ and $\dot{x_{d}}$ are the target end position and the velocity at the target end position. $e=x-x_{d}$ represents the tracking error, and $\dot{e}$ is the the speed error. Additionally, the saturation constraints are imposed on the manipulator joints to constrain the swing amplitude when the manipulators track a target curve. The limitations are as follows:(20)$$q_{i}=\left\{\begin{array}{cc} q_{\min} & ,\;\mathrm{if}\;q_{i}\leq q_{\min}\\ q_{i} & ,\;\mathrm{if}\;q_{\min}<q_{i}<q_{\max},i=1,2,\ldots ,n\\ q_{\max} & ,\;\mathrm{if}\;q_{i}\geq q_{\max} \end{array}\right.$$
where $q_{\max}=\pi$ and $q_{\min}=-\pi$.

In trajectory tracking control of the robot manipulator end effector, the controller’s output signals are the torque acting on the joint angle of the manipulator. The action space is as follows:(21)$$a={\left[\begin{array}{c} \tau \end{array}\right]}^{T}$$
where
(22)$$\tau_{i}=\left\{\begin{array}{cc} \tau_{\min} & ,\;\mathrm{if}\;\tau_{i}\leq \tau_{\min}\\ \tau_{i} & ,\;\mathrm{if}\;\tau_{\min}<\tau_{i}<\tau_{\max},i=1,2,\ldots ,n\\ \tau_{\max} & ,\;\mathrm{if}\;\tau_{i}\geq \tau_{\max} \end{array}\right.$$
where $\tau_{\max}$ is the maximum of the input torque, $\tau_{\min}$ is the minimum of the input torque, and $\tau_{\max}=5$, $\tau_{\min}=-5$.

In RL tasks, the reward is a measure to access the effectiveness of the current policy. In the trajectory tracking task of the manipulator, the tracking error, e(t), is the variable of most concern. Therefore, the reward function is set to a nonlinear form. The reward function is formulated as:(23)$$r=\left(1-\frac{1}{\exp\left(-\varsigma \cdot \left(\bar{e}-\epsilon \right)\right)}\right)$$
where ς denotes a sensitive scale and is set to adjust the rate of increase, enabling the agent to quickly obtain rewards once a certain performance level is reached. ϵ represents a benefit threshold and is used to indicate the bound of rewards [39]. In this paper, ς=2 and ϵ=0.02.


### 4.2. Control Algorithm Based on SAC-LSTM

The actor–critic architecture is employed in most RL algorithms to effectively address policy gradient-based problems. The actor network will eventually train a policy, π, which is the controller required for the tracking task. This policy is refined employing the policy gradient approach under the guidance of the critic network. Concurrently, the critic network needs to be trained to precisely assess the outputs generated by the actor network.

In this paper, the SAC-LSTM employs the actor–critic architecture, including the policy network and critic network, as detailed in Algorithm 1. The network structure is depicted in Figure 3. The actor network includes an LSTM network layer, a fully connected network layer, and an output layer. The state sequence of each time step of the robot manipulator in the trajectory tracking task is used as the input of LSTM, and the hidden state, h, of each time step is used as the output. Then, *h* at each time step in the entire trajectory is used as the input of the fully connected layer. After passing through the fully connected layer network, the Gaussian distribution parameters of the action (torque) are output. During training, the action values are sampled from this distribution, ensuring that the actions of the agent are chosen in a random and exploratory manner. As training progresses, the network converges and the variance decreases gradually. The value network includes a fully connected layer and an output layer, taking the splicing of state sequences and actions as input. The network output layer outputs a single variable, the value of the state sequence, $V_{\phi }\left(S\right)$.
**Algorithm 1** SAC-LSTM**Input:** Policy network parameter ϕ, Q Network parameters $\theta_{1}$, $\theta_{2}$**Input:** Empty replay buffer D**Input:** target network weights ${\hat{\theta }}_{1}\leftarrow \theta_{1},{\bar{\theta }}_{2}\leftarrow \theta_{2}\;$**Output:** Optimized parameters1:**for** each iteration **do**2:    **for** each interaction with environment **do**3:        Get s and choose $a∼\pi_{\phi }\left(.|s_{t}\right)$4:        Perform action a5:        Record $s^{\prime }$, reward r, and6:           termination signal d7:        Save $\left(s,a,r,s^{\prime },d\right)$ in replay buffer D8:        **if** termination signal d is terminal **then**9:           Reset state of environment10:        **end if**11:    **end for**12:    **for** each gradient **do**13:        From D sample a batch of data B=(s, a, r, $s^{\prime }$, d);14:        Calculate targets for the Q networks following (15)15:        Update parameters of Q network according to16:           gradient descent following (16)17:        Update parameters of policy according to18:           gradient descent following (18)19:        Update target network parameters with20:           $\phi_{\mathrm{targ},i}\leftarrow \rho \phi_{\mathrm{targ},i}+\left(1-\rho \right)\phi_{i}\;$ for i=1,221:    **end for**22:**end for**

**Remark 1.** 
*Inspired by the above reference [20], the LSTM network is able to efficiently capture sequence information. It generates current outputs by integrating evolving trends from past data, which is more effective for tasks with periodic changes such as robotic manipulator trajectory tracking. In this paper, retaining the outputs of all time steps from the LSTM and passing them to the fully connected layer offers significant advantages for tracking the trajectory of a robotic manipulator. Firstly, this approach fully utilizes the information from the entire sequence, ensuring that the state at each time step is considered, thereby enhancing the precision of trajectory tracking. Secondly, it helps capture long-term dependencies, enabling the model to better handle complex temporal dependencies. Thirdly, using information from the entire sequence can make the control output smoother, reducing jitter and abrupt changes in the robotic arm’s movement. Additionally, this method improves the model’s robustness when dealing with dynamic environments or uncertain external disturbances, enhancing its ability to adapt and respond to changes.*


### 4.3. GAIL for Robot Manipulator Trajectory Tracking

GAIL is able to learn control policies directly from expert demonstration data, which effectively solve the time-consuming problem of reinforcement learning training. The SAC-LSTM algorithm is used as a generator to train GAIL, for implementing SL-GAIL on robotic manipulators’ trajectory tracking control, as shown in Figure 4. The discriminator of GAIL is composed of a fully connected layer followed by an output layer. Among them, the state–action pair spliced into the state and action is used as the input of the discriminator, and the output is a value between 0 to 1. The robotic manipulator agent interacts with the environment employing the current policy $\pi (a|s;\theta_{n}ow)$, which serves as the generator. This interaction generates a trajectory Ω = [$s_{1},a_{1},s_{2},a_{2},\ldots ,s_{n},a_{n}$] consisting of state–action pairs. For each state–action pair (s,a), the discriminator outputs a value, D(s,a), to determine whether it originates from the expert demonstration or from the policy of the agent. Ideally, D(s,a) should be close to 1 for expert trajectories and close to 0 for agent-generated trajectories. The generator (policy of the agent) is trained employing the SAC-LSTM algorithm to produce state–action pairs that closely match the expert data, thereby fooling the discriminator. The discriminator, conversely, is designed to differentiate between the expert and agent-generated trajectories. Eventually, the generator is trained to maximize $E_{\pi }\left[\log\left(D\left(s,a\right)\right)\right]$ and the discriminator is trained to minimize $E_{\pi }\left[\log\left(D\left(s,a\right)\right)\right]+E_{\pi_{E}}\left[\log\left(1-D\left(s,a\right)\right)\right]$. GAIL uses a surrogate reward:(24)r(s,a)=−log(D(s,a))
This surrogate reward is used to update the policy of the agent. The ultimate objective is learning an optimal control policy that maximizes the expected cumulative discounted return:(25)$$\pi^{*}=\underset{\pi \in \Pi }{\arg\min}E_{\pi }\left[r\left(s,a\right)\right],$$
where $E_{\pi }\left[r\left(s,a\right)\right]≜E\left[\sum_{t}^{T}\gamma^{t}r\left(s,a\right)\right]$ represents an expectation of discounted return according to the trajectory generated by policy π.

This approach, detailed in Algorithm 2, ensures that the policy of the agent evolves to closely mimic expert behavior while efficiently handling the challenges of trajectory tracking control in robotic manipulators.
**Algorithm 2** SL-GAIL**Input:** Expert demonstration data $D_{E}$, Policy network parameter ϕ, Discriminator network parameters ω**Output:** Optimal policy parameters ϕ1:Initialize policy network parameters ϕ2:Initialize discriminator network parameters ω3:**for** each training iteration **do**4:    Sample expert batch $B_{E}$ from $D_{E}$5:    Generate trajectory Ω using current policy π(a|s;ϕ)6:    **for** each state–action pair (s,a) in Ω **do**7:        Compute discriminator output D(s,a)8:        Compute surrogate reward by equation (Equation 24)9:    **end for**10:    Update ϕ using surrogate reward r(s,a)11:    Sample state–action pairs (s,a) from Ω and $B_{E}$12:    Update discriminator parameters ω using equation (Equation 18)13:**end for**

**Remark 2.** 
*Different from the above-mentioned references [24,25], this paper employs the off-policy algorithm SAC as a generator for GAIL. The policy entropy regularization of the SAC algorithm enhances the sampling capability and generates smoother and continuous action outputs, which is more suitable for processing systems such as robotic manipulators. In order to solve the instability of the off-policy algorithm when training the GAIL strategy, the LSTM network is added to the SAC algorithm. LSTM effectively captures the long-term dependencies and dynamic patterns in the sequence data, thereby helping the generator to better predict the continuous action sequence, maintain the temporal coherence of the generated samples, and enhance the model’s adaptability to dynamic changes in the environment, thereby improving the stability of the generator in the GAIL framework.*


## 5. Simulation

In this section, simulation experiments are performed based on the simulation environment in Section 2 to check the capability of the proposed method in this paper. The SAC, SAC-LSTM, and SAC-GAIL algorithms are chosen as the benchmarks to evaluate our SL-GAIL algorithm. Meanwhile, the anti-interference capabilities of the four controllers are evaluated. In this paper, all experiments are examined using the Phantom Omni robotic manipulator. The initial state values are defined as $\theta_{1}\left(0\right)=-0.3$, $\theta_{2}\left(0\right)=0.3$, and $\theta_{3}\left(0\right)=-0.8$. The total steps are 3000, and the step size of each step is 0.01 s.

In the all simulation experiments, $x_{d}$ is selected as
$$x_{d}=\left[\begin{array}{c} x\\ y\\ z \end{array}\right]=\left[\begin{array}{c} 0.1\sin(t)+0.12\\ 0.1\cos(t)+0.12\\ 0.1\sin(t) \end{array}\right]$$
where $t\in (0,t_{limit})$ and $t_{limit}$ = 30 s.

### 5.1. Training Performance

In this experiment, the performance of the SAC-LSTM, SAC-GAIL and SL-GAIL algorithms are tested during the training process. The detailed parameter settings of all algorithms are shown in the Appendix A.

The trends in cumulative reward values obtained by the three algorithms, and the standard deviation and variance of the cumulative reward are shown in Figure 5. The results show that the cumulative rewards value of the SL-GAIL agent tend to converge around the 20th episode and the learning curve is smooth later. However, the SAC-LSTM agent and the SAC-GAIL agent do not converge until after the 100th. At the same time, the variance and standard deviation of SL-GAIL are in a stable state after the 100th episode, but the variance and standard deviation of SAC-LSTM and SAC-GAIL always fluctuate greatly.

Moreover, compared with SAC-GAIL, the SAC-LSTM algorithm is more stable during training. Therefore, when using RL to perform robotic manipulator tracking control tasks, LSTM can enhance the stability of robotic manipulator agent learning and GAIL to improve the efficiency of robotic manipulator agent learning.

### 5.2. Control Performance

The tracking control performance of SL-GAIL is compared with SAC, SAC-LSTM, and SAC-GAIL. Figure 6 shows the tracking trajectories, tracking error, and a change in the torque of the end effector using SAC, SAC-GAIL, SAC-LSTM, and SL-GAIL during the training process in the control task. As demonstrated in Figure 6a–c, the trajectories of SAC, SAC-GAIL, SAC-LSTM, and SL-GAIL agents in the X-, Y-, and Z-axes are shown. The tracking errors of the end effector in the three axes are shown in Figure 6d–f. It can be seen that the SL-GAIL algorithm achieves more superior tracking control performance than the baseline methods. Figure 6g–i show the change trend of the torque for the joint; it can be seen that the torque of the robotic manipulator when controlled by SL-GAIL is smaller than that of SAC, SAC-GAIL, and SAC-LSTM.

As can be seen from this experiment, SL-GAIL combines the strengths of GAIL and LSTM and quickly learns policies that surpass those derived from stable expert data. As a result, it achieves excellent control performance with lower torque requirements. The average value of the absolute value of the tracking error (AAE), the average value of the absolute value of torque (AAT) and root-mean-squared error (RMSE) are shown in Table 2. The following formulas calculate AAE, AAT, and RMSE:(26)$$\mathrm{RMSE}=\sqrt{\frac{1}{T}\sum_{i=1}^{T}e_{i}^{2}}\;i=1,2,\ldots ,n$$
(27)$$\mathrm{AAE}=\frac{1}{T}\sum_{i=1}^{T}\left|e_{i}\right|\;i=1,2,\ldots ,n$$
(28)$$\mathrm{AAT}=\frac{1}{T}\sum_{i=1}^{T}\left|\tau_{i}\right|\;i=1,2,\ldots ,n$$
where *T* is the total time.

### 5.3. Anti-Interference Performance

In this simulation experiment, the proposed method is tested for anti-interfere, $y_{d}$ is the same as in Experiment 1, and the external disturbance, $\tau_{d}$, is selected as:$$\tau_{d}=\left\{\begin{array}{cc} {\left(5\sin(t)\;5\cos(t)\;5\sin(t)\right)}^{T} & a>0.8\\ 0 & \;\mathrm{else}\; \end{array}\right.$$
where *a* represents a random variable uniformly distributed in the interval [0.0, 1.0). If this value exceeds 0.80, a nonlinear disturbance term is added to the system input signal, which means that the perturbation has a twenty percent probability of occurring at each time step. The nonlinear disturbance term is a 3-dimensional column vector composed of the sine and cosine values of the current time step, multiplied by a scaling factor (5). By introducing random disturbances, the robustness of the controller in uncertain environments can be tested. In addition, introducing perturbations in the form of sine and cosine curves can simulate many nonlinear effects in mechanical systems and improve the accuracy of the model.

In order to evaluate the anti-interfere of SL-GAIL, we saved the optimal control policies trained with SAC-LSTM, SAC-GAIL, and SL-GAIL in the curve-following task. The control policies of the three agents are compared with the SL-GAIL algorithm. Figure 7 shows the position tracking of the four algorithms in a disturbance-free environment (DFENV) and a disturbed environment (DENV). The first, second, and third rows are the X, Y, and Z positions of the four algorithms, respectively. It is noted that, compared with the other three algorithms, the position tracking of the SL-GAIL algorithm changes very little in the two environments. The tracking error and torque changes of the four algorithms in the DENV are shown in Figure 8. It can be seen that the torques using SL-GAIL are still smaller than using SAC, SAC-LSTM, and SAC-GAIL. Figure 9 present the AAE and RMSE for all algorithms in the DFENV and DENV. The AAE and RMSE of SL-GAIL tested in the two environments have almost no change. However, the AEE and RMSE of the SAC, SAC-LSTM, and SAC-GAIL change significantly in the two environments. The changes in AAE under the two environments are shown in Table 3. The changes in RMSE under the two environments are shown in Table 4.

From this experiment, it can be seen that, in the robotic arm environment with the same joint angle constraints and maximum torque limits, the anti-interference ability of the SL-GAIL controller proposed in this paper is stronger than that of the three other controllers. It is indicated again that GAIL and LSTM play an important role in this paper in enhancing the robustness of the algorithm.

## 6. Conclusions

In this paper, we addressed the challenge of trajectory tracking in task space for the end effector of the Phantom Omni manipulator. We proposed the SAC-LSTM algorithm to enhance the performance of the control system, particularly in adapting to time-varying trajectories. The integration of long short-term memory (LSTM) allowed the system to effectively capture temporal dependencies in the trajectory, thereby improving the robustness and adaptability of the controller in dynamic environments. By combining SAC-LSTM with generative adversarial imitation learning (GAIL), the SL-GAIL method is able to directly learn from expert demonstrations, significantly improving the efficiency of policy learning and reducing training time. The simulation results demonstrate that the SL-GAIL method not only achieves faster learning but also enhances the robustness of the control system compared with the baseline algorithm. The experimental outcomes highlight the effectiveness of combining reinforcement learning with imitation learning to solve real-world robotic control problems, suggesting that the proposed approach is a promising solution for time-sensitive and high-precision tasks in robotic manipulation.

## Figures and Tables

**Figure 1 biomimetics-09-00779-f001:**
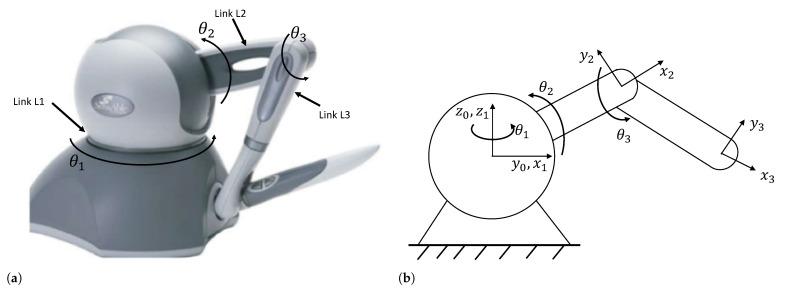
(**a**) The 3-DOF Phantom Omni robot. (**b**) Schematic of the 3-DOF device.

**Figure 2 biomimetics-09-00779-f002:**
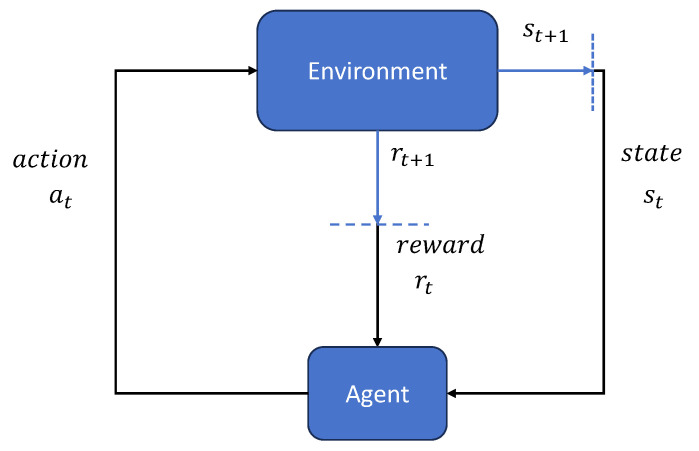
Graph of agent learning with reinforcement learning.

**Figure 3 biomimetics-09-00779-f003:**
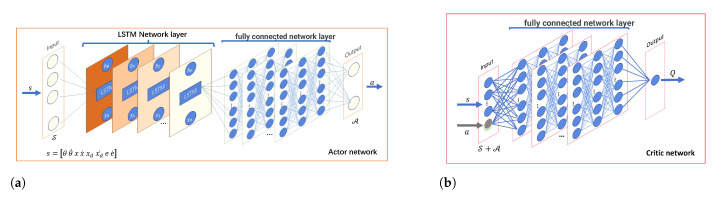
(**a**) Actor network architecture; (**b**) critic network architecture.

**Figure 4 biomimetics-09-00779-f004:**
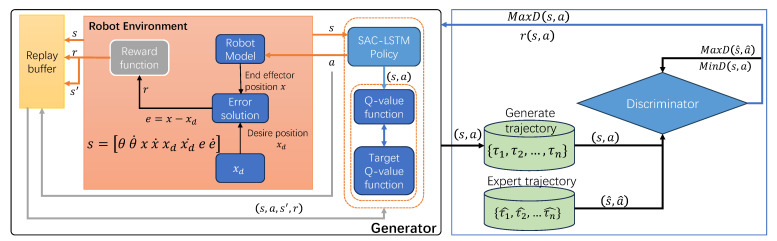
Illustration of the SL-GAIL method.

**Figure 5 biomimetics-09-00779-f005:**
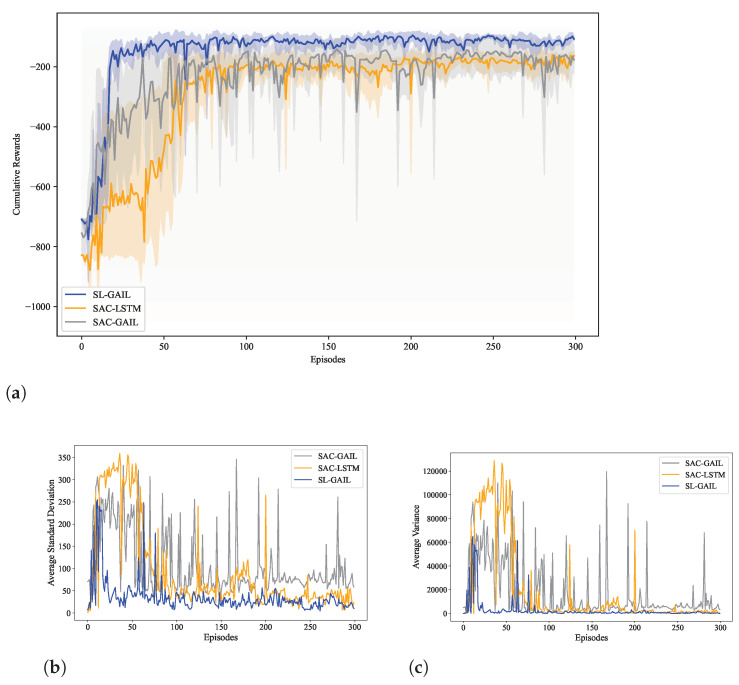
The variance and standard deviation of the cumulative reward value for each episode. (**a**) Cumulative rewards for each episode. (**b**) Standard deviation of cumulative rewards. (**c**) Variance deviation of cumulative rewards.

**Figure 6 biomimetics-09-00779-f006:**
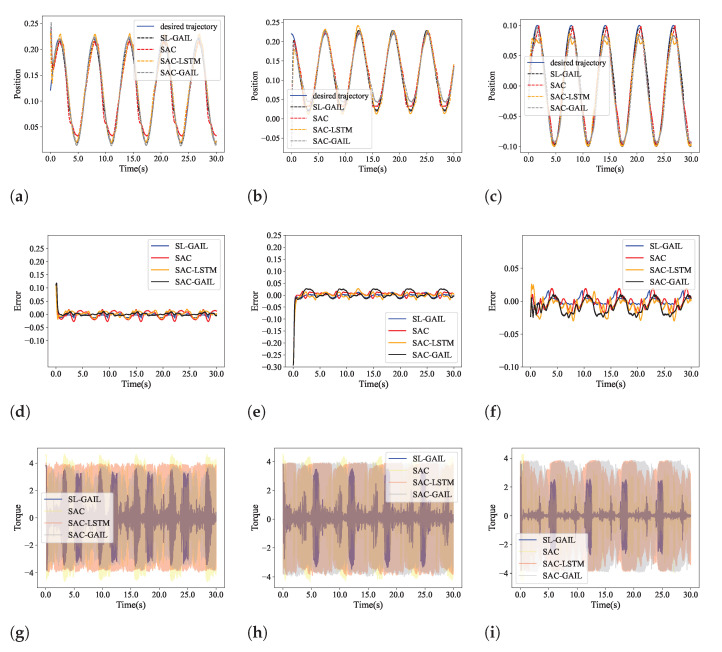
The tracking position, tracking error, and torque in Experiment 1. (**a**–**c**) Tracking position of x-, y-, and z-axes. (**d**–**f**) Tracking error of of x-, y- and z-axes. (**g**–**i**) Torque of the three joints.

**Figure 7 biomimetics-09-00779-f007:**
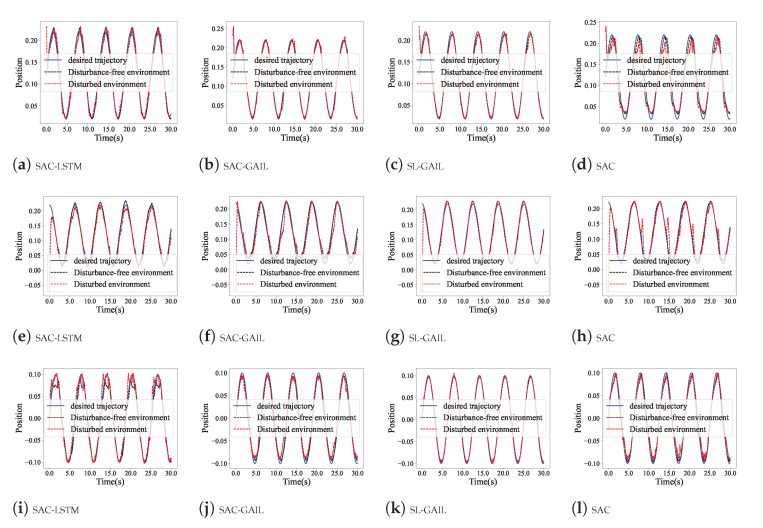
The position tracking in an environment with interference in Experiment 2. (**a**–**d**) Position tracking of x-axes. (**e**–**h**) Position tracking of y-axes. (**i**–**l**) Position tracking of z-axes.

**Figure 8 biomimetics-09-00779-f008:**
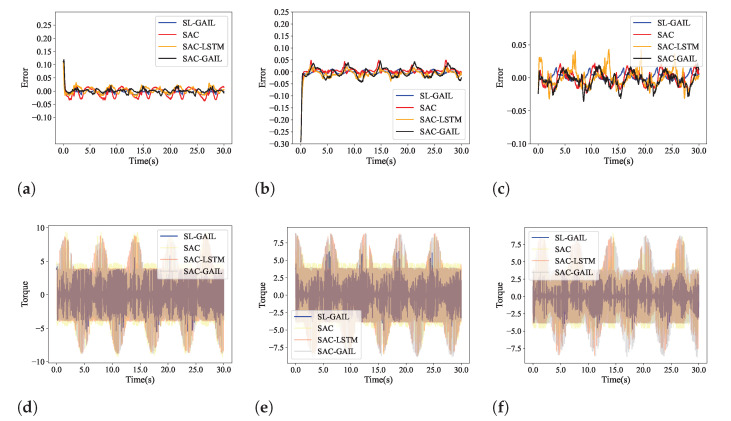
The tracking error and torque in an environment with interference in Experiment 2. (**a**–**c**) tracking error of x-, y- and z-axes. (**d**–**f**) tracking error of the three joints.

**Figure 9 biomimetics-09-00779-f009:**
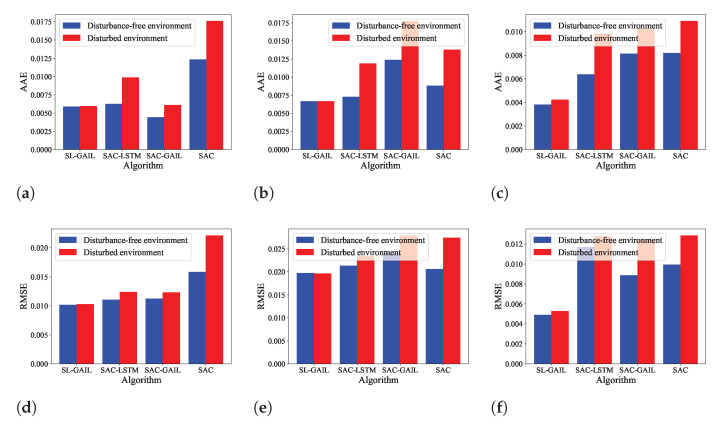
The AAE and RMSE in two environments. (**a**–**c**) The AAE in two environments. (**d**–**f**) The RMSE in two environments.

**Table 1 biomimetics-09-00779-t001:** The controlled objects and their control methods.

Category	Method	Main Constraints	References
Hydraulic Manipulators	RBFNN-based adaptive asymptotic method	Complex mathematical models, uncertain dynamics	[9]
Serial Manipulators	PID, sliding-mode control, adaptive control	High reliance on precise models, many control parameters	[4,5,7]
Free-floating Manipulators	DRL-based SAC-RNN	Unknown dynamics, high control complexity	[19]
3-DoF Manipulators	SARSA algorithm	Weak capability for handling continuous state spaces	[13]
2-DoF Manipulators	Actor–critic RL	Deadzone problem, nonlinear dynamic characteristics	[18]
Pneumatic Musculoskeletal Robots	Model-based RL (MBRL) methods	Difficulty in learning effective control policies	[14]
Autonomous Underwater Vehicles (AUVs)	GAIL-based policy learning	Low data efficiency, challenging generalization to unknown environments	[24]
Unmanned Surface Vehicles (USVs)	GAIL combined with PPO or TRPO	Low sample efficiency, insufficient exploration capability	[25]
Robotic Arms and Mobile Robots	Distributed PPO combined with LSTM	Insufficient handling of temporal state dependencies	[16]
Underactuated Autonomous Surface/Underwater Vehicles (UASUVs)	SAC combined with LSTM	Training instability, input disturbances	[20]

**Table 2 biomimetics-09-00779-t002:** AAE, AAT, and RMSE of SL-GAIL, SAC-LSTM, and SAC-GAIL algorithms in training.

Axe Algorithms		SAC	SAC-GAIL	SAC-LSTM	SL-GAIL
X	AAE	0.0123	0.0044	0.0062	0.0058
	AAT	3.5349	1.9642	2.2820	1.1057
	RMSE	0.0158	0.0112	0.0110	0.0101
Y	AAE	0.0088	0.0123	0.0072	0.0066
	AAT	3.4609	2.6186	1.8884	0.7469
	RMSE	0.0206	0.0240	0.0213	0.01975
Z	AAE	0.0081	0.0081	0.0063	0.0038
	AAT	2.9630	2.3313	0.9714	0.4388
	RMSE	0.0099	0.0088	0.0116	0.0052

**Table 3 biomimetics-09-00779-t003:** Comparison of AAE in two environments.

Axes Algorithms		SAC	SAC-GAIL	SAC-LSTM	SL-GAIL
X	DFENV	0.0123	0.0044	0.0062	0.0058
	DENV	0.0176	0.0061	0.0098	0.0059
Y	DFENV	0.0088	0.0123	0.0072	0.0066
	DENV	0.0137	0.0177	0.0118	0.0066
Z	DFENV	0.0081	0.0081	0.0063	0.0038
	DENV	0.0099	0.0102	0.0098	0.0042

**Table 4 biomimetics-09-00779-t004:** Comparison of RMSE in two environments.

Axes Algorithms		SAC	SAC-GAIL	SAC-LSTM	SL-GAIL
X	DFENV	0.0158	0.0112	0.0110	0.0101
	DENV	0.0221	0.0123	0.0124	0.0102
Y	DFENV	0.0206	0.0240	0.0213	0.0197
	DENV	0.0274	0.0279	0.0234	0.0196
Z	DFENV	0.0099	0.0088	0.0116	0.0049
	DENV	0.0128	0.0124	0.0127	0.0052

## Data Availability

Data are contained within the article.

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
