# Peer review of "Trajectory Tracking Control for Robotic Manipulator Based on Soft Actor–Critic and Generative Adversarial Imitation Learning"

_biomimetics, 2024, doi:10.3390/biomimetics9120779_

Round 1
Reviewer 1 Report
Comments and Suggestions for Authors
This paper proposes a deep reinforcement learning (DRL) approach using generative adversarial limitation learning (GAIL) and long short-term memory (LSTM) to solve the robotic manipulator tracking control problem with saturation constraints and random disturbances without learning the manipulator's dynamic and kinematic model. However there are following concerns:
1. In Abstract, the author is proposing his/her algorithm to address tracking control problem for robotic manipulator with saturation constraints and random disturbances- which has been addressed by several researchers. Thus, it is suggested to highlight the main constraints or limitations in the abstract which your algorithm is going to address. I am unable to find any statement within the abstract that shares the novelty of the work.
2. The literature review has been presented well. Small suggestion is to form a table that shares the categories of robotic manipulators and their types along with major constraints. In order to add this, please read and must cite below papers:
a. Abdelmaksoud, Sherif I., Mohammed H. Al-Mola, Ghulam E. Mustafa Abro, and Vijanth Sagayan Asirvadam. "In-Depth Review of Advanced Control Strategies and Cutting-Edge Trends in Robot Manipulators: Analyzing the Latest Developments and Techniques." IEEE Access (2024).
b. Pan, Jinghui. "Fractional-order Sliding Mode Control of Manipulator Combined with Disturbance and State Observer." Robotics and Autonomous Systems (2024): 104840.
Furthermore, Identify which robotic manipulator has been used by the author for this paper and how its related limitation have been addressed.
3. Since the paper is proposing new techniques, I suggest to come up with the table that shares all state of the art approaches and their respective limitations.
4. Regarding the equations, some of the variables are not defined like description is missing. Therefore I suggest you to add one table sharing all symbols and their description as well.
5. Conclusion is not written appropriately. Please add on more exhuasive information in it.
Comments on the Quality of English LanguageRe-read the entire paper to avoid grammatical and punctuation errors and avoid using pronouns i.e. I, we and they in entire paper
Author Response
Comments 1: In Abstract, the author is proposing his/her algorithm to address tracking control problem for robotic manipulator with saturation constraints and random disturbances- which has been addressed by several researchers. Thus, it is suggested to highlight the main constraints or limitations in the abstract which your algorithm is going to address. I am unable to find any statement within the abstract that shares the novelty of the work. |
Response 1: Thank you for pointing this out. We agree with this comment. The main innovation of our paper is in the algorithm, so we omitted the description of the constraints and limitations of the robot. We have made changes to the abstract as follows: In this paper, a deep reinforcement learning(DRL) approach based on generative adversarial imitation learning(GAIL) and long short-term memory(LSTM) is proposed to resolve tracking control problem for robotic manipulator with saturation constraints and random disturbances, without learning the dynamic and kinematic model of the manipulator. Specifically, it limits the torque and joint angle to a certain range. Firstly,.... Secondly, .... Finally, .... |
Comments 2: The literature review has been presented well. Small suggestion is to form a table that shares the categories of robotic manipulators and their types along with major constraints. In order to add this, please read and must cite below papers: a. Abdelmaksoud, Sherif I., Mohammed H. Al-Mola, Ghulam E. Mustafa Abro, and Vijanth Sagayan Asirvadam. "In-Depth Review of Advanced Control Strategies and Cutting-Edge Trends in Robot Manipulators: Analyzing the Latest Developments and Techniques." IEEE Access (2024). b. Pan, Jinghui. "Fractional-order Sliding Mode Control of Manipulator Combined with Disturbance and State Observer." Robotics and Autonomous Systems (2024): 104840. Furthermore, Identify which robotic manipulator has been used by the author for this paper and how its related limitation have been addressed. |
Response 2: Agree. We have, accordingly, done a table to emphasize this point. We carefully read the two articles recommended by the reviewers, which greatly helped to enrich the content of our article, so we cited them in the article. In addition, we listed a table to show the control objects and controllers used in the articles mentioned in the literature review section. The specific table is shown in Table 1 in the manuscript. In addition, the robot manipulator we used is shown in detail in the system description in Chapter 2. As for how to solve the problem of its related limitations, the control method we adopted is deep reinforcement learning, which regards the manipulator as an intelligent agent and actively learns the control policy. This is a learning process similar to a black box.
|
Comments 3: Since the paper is proposing new techniques, I suggest to come up with the table that shares all state of the art approaches and their respective limitations. |
Response 3:Table 1 in the manuscript lists the control of objects such as robotic arms by deep reinforcement learning in recent years. |
Comments 4: Regarding the equations, some of the variables are not defined like description is missing. Therefore I suggest you to add one table sharing all symbols and their description as well. |
Response 4:Thanks for pointing this out.Table A3 in the Appendix shows the definitions of some variables. |
Comments 5: Conclusion is not written appropriately. Please add on more exhuasive information in it. |
Response 5:Thank you for pointing this out. We agree with this comment. Please see the conclusion section of the manuscript for detailed revisions |
Please see the attachment.

Reviewer 2 Report
Comments and Suggestions for Authors
The paper deals with a new, advanced method for designing a controller for the task of trajectory tracking control. For this purpose, an approach was developed that combines modern methods of artificial intelligence: Soft Actor-Critic and Generative Adversarial Imitation Learning.
The considerations made are characterized by great attention to detail.
The aim of the work and the results achieved were presented very clearly.
The results of numerous simulation studies confirming the effectiveness of the new method were also presented.
I would like to draw attention to one aspect of the work that should perhaps be reconsidered. In equation (19), both the classical state variables and their derivatives are given as state variables. Why was it decided to take such a step? In my opinion, this causes confusion. It should therefore be explained exactly why x and \dot{x} are treated as separate state variables.
I rate the work highly. I believe that the paper can be accepted for publication. Nevertheless, I would encourage you to consider the above problem.
Author Response
Comments 1: The considerations made are characterized by great attention to detail. The aim of the work and the results achieved were presented very clearly. The results of numerous simulation studies confirming the effectiveness of the new method were also presented. |
Response 1: Thank you for your positive feedback. I really appreciate the attention to detail in this article. Your comments motivate me to continue improving and enhancing the quality of my work. |
Comments 2: I would like to draw attention to one aspect of the work that should perhaps be reconsidered. In equation (19), both the classical state variables and their derivatives are given as state variables. Why was it decided to take such a step? In my opinion, this causes confusion. It should therefore be explained exactly why x and \dot{x} are treated as separate state variables. |
Response 2: Thanks to the reviewer for his comments. The traditional state space includes joint angles and joint angular velocities, which can describe the internal state of the robot arm, but cannot directly reflect the relationship between the performance of the end effector of the robot arm and the target. Therefore, adding the mapped end point position and velocity, as well as the target position and velocity, can provide more relevant information about the end effector and the target trajectory, thereby helping the reinforcement learning algorithm to control more accurately. We incorporate these variables into the state space in order to provide the algorithm with more comprehensive environmental feedback, thereby accelerating the learning process and improving the robustness of the model. The detailed changes in the manuscript are from lines 218 to 224 on page 7. |
Please see the attachment.
